# Metallic porous nitride single crystals at two-centimeter scale delivering enhanced pseudocapacitance

Shaobo Xi[1], Guoming Lin[1], Lu Jin[1], Hao Li[1] & Kui Xie [1]*

Pseudocapacitors that originate from chemisorption contain redox active sites mainly composed of transition metal ions with unsaturated coordination in lattice on the electrode surface. The capacitance is generally dictated by the synergy of the porous microstructure, electronic conduction and active sites in the porous electrode. Here we grow metallic porous nitride single crystals at 2-cm scale to enhance pseudocapacitance through the combination of large surface area with porous microstructure, high conductivity with metallic states and ordered active sites with unsaturated coordination at twisted surfaces. We show the enhanced gravimetric and areal pseudocapacitance and excellent cycling stability both in acidic and alkaline electrolyte with porous MoN, $Ta_5N_6$ and TiN single crystals. The long-range ordering of active metal-nitrogen sites account for the fast redox reactions in chemisorption while the high conductivity together with porous microstructure facilitate the charge transfer and species diffusion in electrodes.

[1] CAS Key Laboratory of Design and Assembly of Functional Nanostructures, and Fujian Provincial Key Lab of Nanomaterials, Fujian Institute of Research on the Structure of Matter, Chinese Academy of Sciences, 350002 Fuzhou, Fujian, China. *email: kxie@fjirsm.ac.cn

Pseudocapacitance originates from chemisorption regions or ion insertion during electrochemical sweeping[1–3]. Pseudo-capacitors that originate from redox reactions conventionally use nonstoichiometric metal-oxide electrodes and these materials are always facing the challenges of low conductivity and chemical stability while remarkable capacitance reduction is also present due to their irreversibility during the redox reactions. Ruthenium dioxide is a typical oxide electrode which has a high capacitance, reversible charge-discharge features, and significant electrical conductivity, all of which make it highly promising to achieve higher capacitance. However, the high cost makes it unsuitable for many applications. Transition metal oxides like $MnO_2$ have attracted increasing attention; however, the low conductivity and chemical stability of these materials remain fundamental challenges.

Transition-metal nitrides with unsaturated nitrogen coordination are generally a kind of electronic conductors. The nearest neighboring metal ions with short distances would provide a possibility of formation of metallic bond that facilitates electron transport in the connected pathway[4–9]. Coordinatively unsaturated metal-nitrogen active sites on electrode surfaces normally offer the ability to host redox reactions that give rise to reversible energy storage in the form of pseudocapacitance. The local structures of transition metal ions with unsaturated nitrogen coordination would accommodate the reversible chemisorption of specific ions in electrolyte and therefore lead to the redox reactions on electrode surfaces[10–13]. Porous nitride materials could provide a large surface area to host chemisorption while the inside interconnected pores and channels would facilitate the ion diffusion in electrolyte. Porous nitride with exposed unsaturated metal-nitrogen active sites would therefore be highly anticipated to deliver the synergy of porous microstructure, electronic conduction and active sites in electrode for higher energy density.

Single-crystalline metallic porous nitride would provide the possibility to well fit these rigorous requirements. The structural coherence in single crystals would confine the ordered active sites with unsaturated nitrogen coordination in lattice, which hence creates the electrochemically active surface to host fast redox reactions in chemisorption[7,8]. The utmost atom termination layer is clear at atomic scale on the surface of nitride single crystals and may be another advantage to create active surfaces through tailoring the crystal surface structures. Porosity in crystals would create large surface areas while the interconnected pores would make them highly accessible in electrolyte, which would considerably facilitate the species diffusions and fast redox reactions in porous electrodes. Porous single-crystalline nitrides with unsaturated nitrogen coordination are generally a kind of electronic conductor, and some of them have been confirmed to be chemically stable both in acidic and alkaline electrolyte[7].

Here we grow metallic porous transition-metal nitride single crystals at a 2-cm scale to enhance pseudocapacitance by a combination of long-range ordering of active sites on a twisted surface, electronic conduction of metallic states and large surface areas of porous microstructures. We show the enhanced gravimetric and areal pseudocapacitance and excellent cycling stability with a metallic porous nitride single crystal. We disclose the fast redox reactions in the reversible chemisorption on metal-nitrogen active sites which are composed of transition metal ions with unsaturated nitrogen coordination numbers.

## Results

**Crystal growth.** In this work, we grow metallic porous MoN single crystals through a lattice reconstruction strategy by direct conversion of $PbMoO_4$ single crystals in a vacuumed ammonia atmosphere at high temperature. Figure 1a shows the growth mechanism of MoN crystal in relation to different lattice channels in $PbMoO_4$ crystals. The [001], [100], and [110] facets of $PbMoO_4$ crystals show suitable lattice channels for the diffusion of atomic lead[14]. Upon evaporation of Pb/O from lattice, the meta-stable Mo–O framework is formed and simultaneously nitrided in ammonia by the substitution of O with N, which leads to the growth of metallic MoN single crystals and therefore maintains a porosity of ~60% in the porous microstructures. Figure 1b shows the XRD patterns of parent $PbMoO_4$ single crystals with facets of [001], [100], and [110], while inset images show the corresponding optical images of polished $PbMoO_4$ parent crystals[15]. Figure 1c shows the XRD pattern of porous MoN single crystal with <001> orientation which is directly grown from the parent $PbMoO_4$ crystals even with three different facets of [001], [100], and [110] under the identical conditions. The [001] facet of MoN is the most thermodynamically stable facet, which therefore leads to the preferential growth of <001> orientation in ammonia at high temperature. We further conduct the measurement of rocking curves which give full width at half max (FWHM) as small as 0.179, 0.193, and 0.092° for the porous MoN, $Ta_5N_6$, and TiN single crystals, respectively, as shown in Supplementary Fig. 1. The porous MoN single crystal with dimensions of $20 \times 10 \times 0.5$ mm is harvested and shown in the inset image. No residual oxygen is observed while the mole ratio between N and Mo is at 1.0 in the porous MoN single crystals according to our element analysis as shown in the EDX mapping in Supplementary Fig. 2. Figure 1d shows the microstructure of MoN single crystal with <001> orientation while the porosity can be further tailored by changing atmospheric pressure as shown in Supplementary Fig. 3. The interconnected pores maintain open frameworks, which therefore combines the advantage of structural coherence and porous microstructures. Metallic porous TiN and $Ta_5N_6$ single crystals are also grown through lattice reconstruction strategy, respectively, as shown in Supplementary Figs. 4 and 5. The specific surface areas are ~14 m² g⁻¹ with the average pore size of ~40 nm for porous MoN single crystals as shown in Supplementary Fig. 6, which indicates the homogeneous distribution of porous microstructures even with different thickness. The porous $Ta_5N_6$ single crystal gives ~2 m² g⁻¹ for specific surface area and ~40 nm for the average pore size while the porous TiN single crystal gives ~6 m² g⁻¹ for specific surface area and ~100 nm for the average pore size.

The microstructures of porous MoN single crystals are examined using transmission electron microscopy (TEM) coupled with focused ion beam (FIB). Figure 2e shows the cross-sectional view of the porous MoN crystal grown along the <001> direction of $PbMoO_4$ parent crystal, which confirms the well-interconnected pores with the diameter of 40–50 nm. The selected area electron diffraction (SAED) at different locations further confirms the identical facet orientation, as well as the single-crystalline nature with hexagonal phase in Fig. 2a–d, f–i. The metallic porous MoN single crystal grown with [100] $PbMoO_4$ shows similar microstructure as shown in Supplementary Fig. 7, in which the identical single-crystalline nature is also observed even at different locations. Both porosity and microstructure are similar for the porous MoN single crystals even though they are grown along different facet directions of $PbMoO_4$ crystals. We use spherical aberration corrected scanning transmission electron microscope (Cs-corrected STEM) coupled with focused ion beam (FIB) to further investigate the nature of porous MoN crystal. Figure 3a shows the high-resolution Cs-corrected STEM image of porous MoN crystal and the lattice spacing of 0.561 and 0.248 nm are ascribed to the [001] and [020] lattice fringes of MoN, respectively. The single-crystalline structure is further confirmed by SAED pattern in the inset of Fig.3a. Details

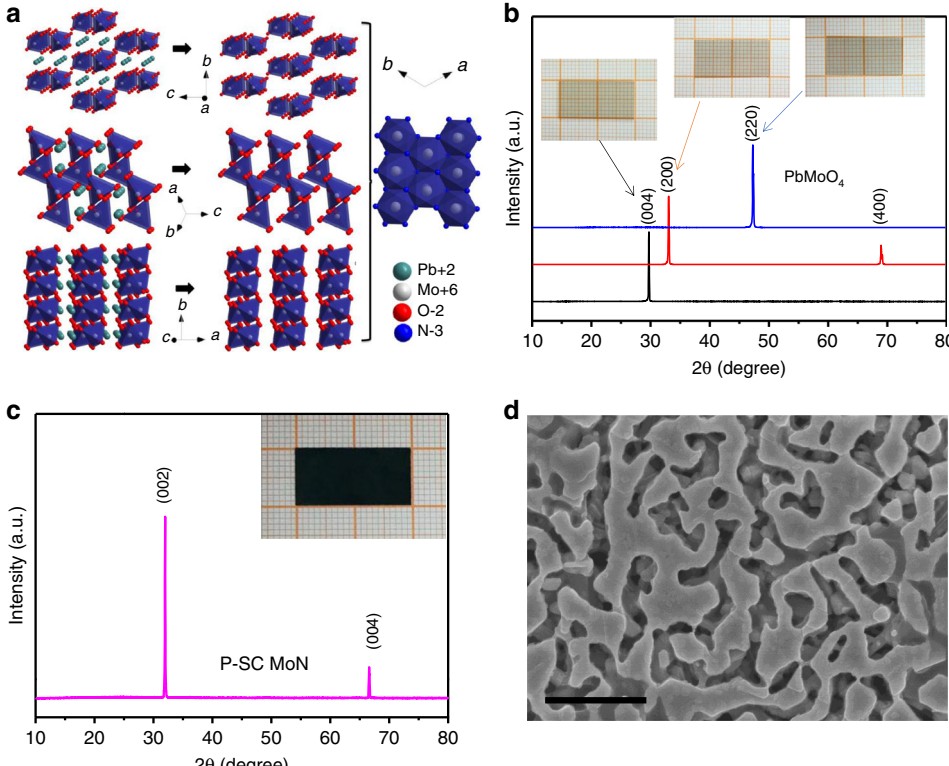

**Fig. 1** Growth mechanism and crystal structure. **a** The growth mechanism of MoN through lattice reconstruction strategy from $PbMoO_4$ crystals. **b** The XRD patterns of $PbMoO_4$ with [001], [100], and [110] facets, the digital images are the parent crystals with dimensions of 20 × 10 × 0.5 mm. **c** The XRD pattern of MoN grown from $PbMoO_4$ crystals, the digital images is the metallic porous MoN single crystals with dimensions of 20 × 10 × 0.5 mm. **d** The porous microstructure of MoN single crystal. The scale bar is 500 nm in panel **d**

of partial enlargement of Cs-corrected STEM image and the corresponding atom distribution are shown in Fig. 3b, which further approves the single-crystalline features of porous MoN crystal in hexagonal phase.

**Active sites**. We use high-sensitive low energy ion scattering (HS-LEIS) with $He^+$ (3 keV) and $Ne^+$ (5 keV) ions sources to detect the outmost atomic surface of porous MoN single crystals. In Fig. 3c, both $He^+$ and $Ne^+$ ion scattering spectra show the presence of atomic Mo termination layer, while the N atoms are beneath Mo atom layer as confirmed in the $He^+$ spectrum. The trace of O atoms signal is attributed to the adsorbed oxygen on surface. The ordered triangle $Mo-N_{1/2}$ structure would function as active metal-nitrogen sites to host fast redox reactions in chemisorption on surface. X-ray photoelectron spectra (XPS) show the Mo–N interactions with a binding energy of 397.82 eV for N1s and two binding energies at 228.79 and 231.92 eV for Mo $3d_{5/2}$ and Mo $3d_{3/2}$ in Mo–N bonds, respectively[16–19]. Figure 3d shows the simulated charge density of ordered active $Mo-N_{1/2}$ sites with partial charge transferring from Mo to N atoms that delivers strong Mo–N interactions. The atomic termination layer of Ta and Ti are also confirmed for the porous $Ta_5N_6$ and TiN single crystals as shown in Supplementary Figs. 4 and 5. The long-range ordering of active sites at twisted surfaces would deliver highly-accessible electrochemically active surfaces to host fast redox reactions in porous microstructures. We summarize the surface characteristics of the three porous single crystals as shown in Supplementary Fig. 8. The porous MoN, TiN and $Ta_5N_6$ single crystals atomically terminate with Mo, Ti and Ta atom layer, respectively, with nitrogen atom layer stacking beneath the surface metal atom layer. The ordered active sites are

confirmed to be $Mo-N_{1/2}$, $Ti-N_{1/2}$, and $Ta-N_{3/5}$ which give rise to electrochemically active surfaces to host fast redox reactions in chemisorption.

The pseudocapacitance originates from the fast redox reactions through species chemisorption which would be dominated by the ordered active sites composed of transition metal ions with unsaturated nitrogen coordination. According to the charge balance, only pseudocapacitance with $OH^-$ adsorption would be observed for the porous TiN single crystal while pseudocapacitance both with $OH^-$ and $H^+$ adsorption would be observed for porous MoN and $Ta_5N_6$ single crystals. To get atomistic insight to chemisorption on ordered active sites, density functional theory (DFT) calculation is conducted to investigate the adsorption energy of specific species on crystal surface both in acidic and alkaline electrolytes. In our work, we consider the $H^+$ in acidic electrolyte and $OH^-$ in alkaline electrolyte as the adsorbed species when we calculate the adsorption energy of species on the surface of porous single crystals[20–22]. We have compared the calculated parameters with theoretical and experimental values as listed in Supplementary Table 1[23–28]. The computed values of MoN, TiN, and $Ta_5N_6$ are well consistent with the theoretical and experimental values, which further indicates the reliability of theoretical calculation in our work. We further conduct the calculation of the band gaps and electronic structures for the porous MoN, TiN, and $Ta_5N_6$ single crystals. These nitrides are typical electronic conductors with band gaps of 0 eV as shown in Supplementary Fig. 9, which further validates the metallic properties of these porous nitride single crystals. We can clearly see from the density of states (DOS) map that the valence band is mainly composed of 2s and 2p orbital of N atoms and 3d orbital transition metal atoms. The conduction band is mainly

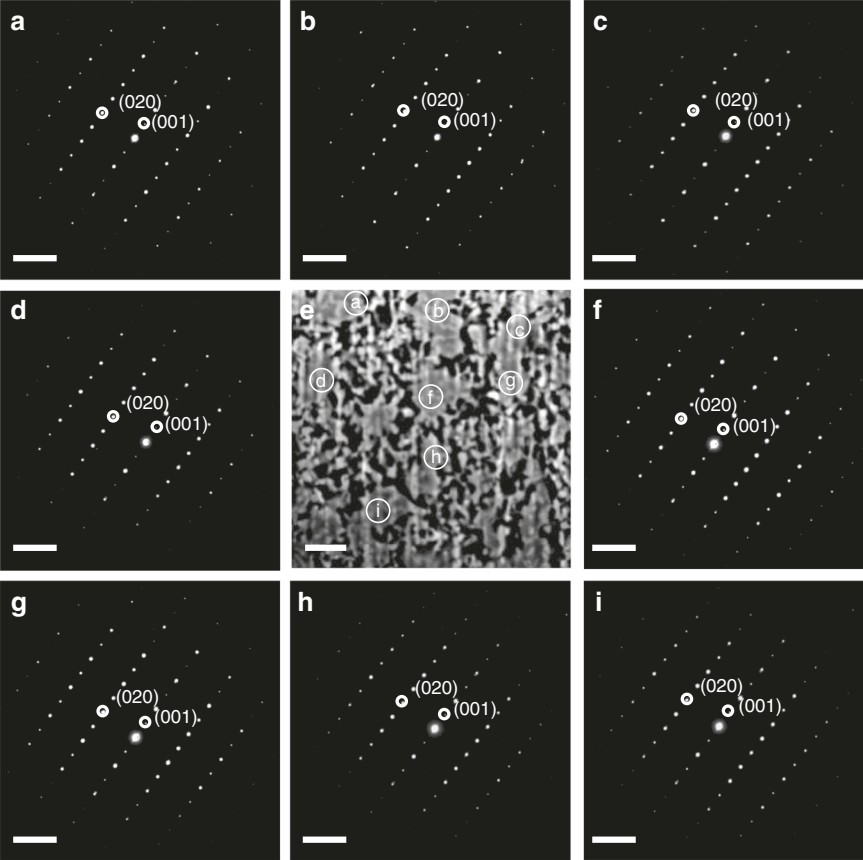

**Fig. 2** Cross-sectional view and selected area electron diffraction. The **a–d** and **f–i** present the SAED pattern at different locations on the cross-section of the porous single-crystalline MoN. The **e** represents the cross-sectional view of the porous single-crystalline MoN with the locations for SAED patterns labeled. The data is recorded with the porous single-crystalline MoN grown on the [001] PbMoO$_4$ substrate. The scale bar is 5 1/nm in **a–d** and **f–i** while the scale bar is 1 μm in panel **e**

dominated by the 3d orbital of transition metal atoms while the 2p orbital of N atoms also has a small contribution.

**Chemical adsorption**. Figure 4a shows the adsorption energy of H$^+$ on [001] MoN surface, where the H binds with the nearest three Mo atoms with bond lengths in the range of 1.97–2.01 Å, indicating a strong chemisorption with a redox reaction. The adsorption energies range from approximately −0.7 to −0.1 eV for the different adsorption configurations which confirms the effective chemisorption in acidic electrolyte. Figure 4b shows the adsorption energies of OH$^−$ on [001] MoN surface, where the O binds with the nearest three Mo atoms with bond lengths in the range of 2.18–2.22 Å. The chemisorption of OH$^−$ leads to the partial oxidation of Mo in Mo–N local structures, showing the adsorption energies of approximately −1.63 to −1.82 eV for different adsorption configurations. The higher adsorption energy of OH$^−$ than H$^+$ on MoN surface would deliver higher pseudocapacitance in alkaline electrolyte in contrast to the performance in acidic electrolyte, which well fits the phenomena observed in our experimental results both in KOH and H$_2$SO$_4$ electrolytes. Actually, the chemisorption is only a necessary condition while the higher adsorption energy indicates the higher energy barrier of desorption on surface. Reasonable chemisorption energy would be favorable to the enhanced pseudocapacitance with porous single crystals.

Figure 4c shows the chemisorption of OH$^−$ on [001] Ta$_5$N$_6$ surface, where the O atom binds with the nearest three Ta atoms, leading to the partial oxidation of Ta in Ta–N local structures.

The adsorption energies of OH$^−$ are in the range of approximately −2.19 to −2.56 eV for different chemisorption configurations. In contrast, the chemisorption energies of H$^+$ on [001] Ta$_5$N$_6$ surface are as high as approximately −1.10 to −1.32 eV for different configurations as shown in Supplementary Fig. 10. Higher adsorption energies give rise to enhanced pseudocapacitance with Ta$_5$N$_6$ in acidic electrolyte than that in alkaline electrolyte, which is well consistent with our experimental results. Figure 4d shows the chemisorption of OH$^−$ on [100] TiN surface where the nearest three Ti atom bind with three O atoms, leading to the bond lengths of 1.89–2.24 Å with the adsorption energy of approximately −0.55 to −1.0 eV. The chemisorption of OH$^−$ is indeed a process of partial oxidation of Ti in Ti–N local structures that results in a proper pseudocapacitance in alkaline electrolyte. However, the adsorption energy of H$^+$ on [100] TiN surface is an endothermic process with positive adsorption energies, which well fits the negligible pseudocapacitance with porous TiN single crystal as observed in acidic electrolyte. We further summarize the adsorption energies of H$^+$ and OH$^−$ on similar surfaces in reported work and list them in Supplementary Table 2[29–36]. The adsorption energy is normally below 0 which indicates an exothermic process that favors the chemisorption on different surfaces of metallic substrates. The calculated adsorption energy in our work is well consistent with the reported values. The chemisorption of specific species like H$^+$ and OH$^−$ on ordered active sites would lead to redox reactions that deliver pseudocapacitance, while reasonably high chemisorption energy greatly would favor the enhancement of pseudocapacitance.

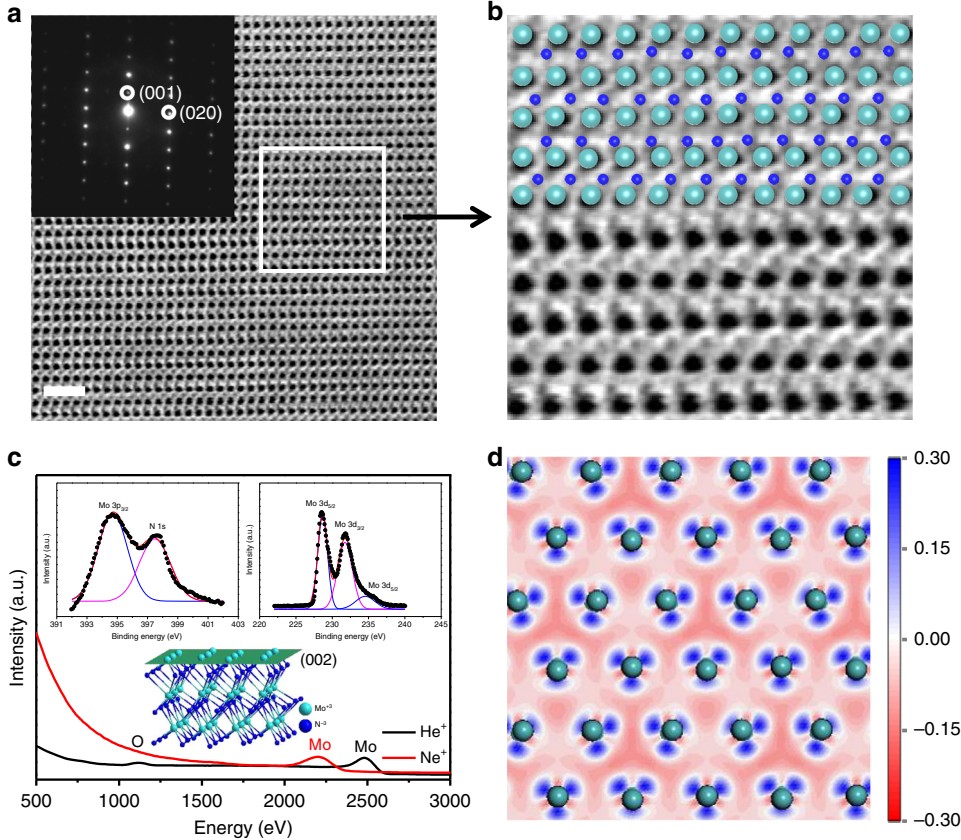

**Fig. 3** Crystal structure and chemical state. **a** Cs-corrected STEM image of the porous single-crystalline MoN. Inset: corresponding SAED pattern of the single-crystalline MoN. **b** Enlargement of partial Cs-corrected STEM image and the corresponding Mo and N atoms distribution. **c** HS-LEIS spectra of the outmost surface layer of single-crystalline MoN. Inset: the structure of MoN showing the [002] plane. **d** Simulated charge density graph of the outmost atomic surface layer of single-crystalline MoN. The dark-blue and light-blue balls represent Mo and N atoms, respectively. The scale bar is 1 nm in panel **a**

**Pseudocapacitor performance**. These transition-metal nitrides are electronic conductors with conductivity as high as $0.77 \times 10^4$ for MoN, $0.8 \times 10^4$ for TiN, and $0.45 \times 10^4$ S cm$^{-1}$ for Ta$_5$N$_6$ single crystals as shown in Supplementary Fig. 11. All these three porous single crystals demonstrate metallic conduction behaviors with the highest conductivity observed at the temperatures of ~ 50 K while the conductivity is decreased by ~50% at room temperature. Figure 5a shows the CV curve with porous MoN single crystal electrode in 1 M KOH solution. The nearly rectangular CV curve indicates the good capacitance property, which may be due to the excellent charge propagation of electrons in the metallic porous MoN single crystals. The triangular CD curves in Fig. 5b shows the exceptional symmetrical characteristic of charge-discharge curves in alkaline electrolyte, which further reveals that the metallic porous MoN single crystals have excellent electrochemical capability and reversibility. In Fig. 5c, the bulk solution resistance ($R_s$) is estimated to be 1.6 Ω while the absence of an impedance arc in the high-frequency region indicates a very low resistivity of MoN single crystal in aqueous alkali solution, due to the synergy of highly accessible surface area, intrinsic electrical conductivity and electrochemical activity. Figure 5d shows the exceptionally high stability of cycling performance with no degradation being observed even after 10,000 cycles. We believe that there may be two reasons for the excellent cycling stability. Firstly, these transition metal nitrides are chemically stable both in acid and alkaline solutions even in the presence of chemical adsorption of H$^+$ or OH$^-$ on surfaces. Secondly, these porous single crystals, free of grain boundary, have the unique advantage of structural coherence, which reduces the interface

defects at the largest extent and hence enhances the stability. Figure 5e shows that the current density is linearly changed versus the scan rate, indicating that the pseudocapacitance is mainly dominated by the chemisorption of the OH$^-$ and H$^+$ on ordered active sites on the surface of porous MoN single crystal.

The chemisorption of OH$^-$ ions is an oxidation process of Mo in ordered local structures, which therefore delivers the exceptionally high pseudocapacitance in alkaline electrolyte. We further conduct Electron Paramagnetic Resonance (EPR) to analyze the porous MoN single crystal which shows EPR signals with g factor of 2.27 that originate from the free electrons in the coordinatively unsaturated Mo–N local structures[37]. We then summarize the unsaturated metal-nitrogen active sites and adsorption energies as shown in Supplementary Fig. 12. The metal-nitrogen coordination structure would function as active sites to accommodate the fast redox reactions in chemisorption on electrode surfaces, which therefore gives the enhanced pseudocapacitance with metallic porous single crystals. Similarly, enhanced pseudocapacitance performance and cycling stability are also observed in acidic electrolyte, which is mainly originating from the chemisorption of H$^+$ that leads to the reduction of Mo in ordered local structures as shown in Supplementary Fig. 13. Improved pseudocapacitance performance is observed for Ta$_5$N$_6$ electrode with reversible chemisorption both in alkaline and acidic electrolyte as shown in Supplementary Figs. 14 and 15. It should be noted that only oxidation reactions with chemisorption of OH$^-$ on the surface of porous TiN single crystal electrode leads to pseudocapacitance in alkaline electrolyte in Supplementary Fig. 16 while the chemisorption of H$^+$ that

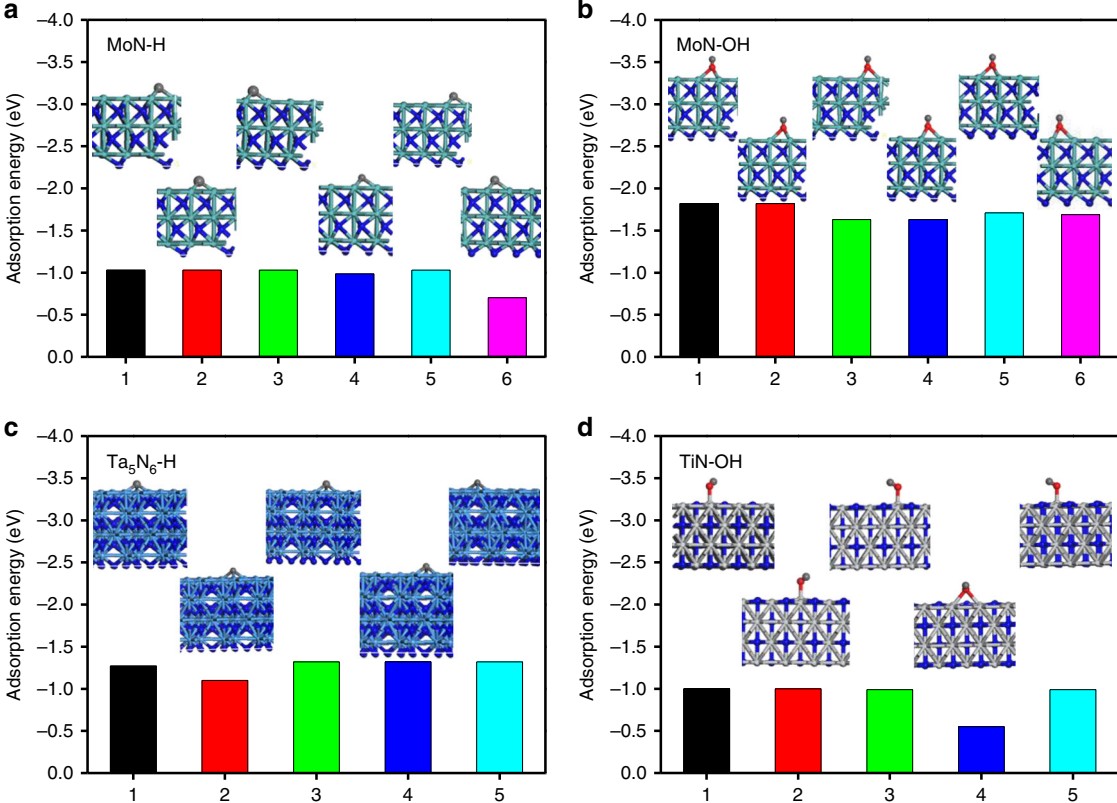

**Fig. 4** The calculated adsorption energy of species on crystal surface. The **a** $H^+$ and **b** $OH^-$ adsorption on MoN surface. **c** The $H^+$ adsorption on $Ta_5N_6$ surface. **d** The $OH^-$ adsorption on TiN surface. Inset images are the corresponding adsorption configurations

results in reduction of Ti is not favorable and contributes negligible pseudocapacitance in acidic electrolyte in Supplementary Fig. 17.

**Pseudocapacitance mechanism.** The linear dependence of capacitance performance on the current densities and sweep rates clearly confirms the typical characteristics of pseudocapacitance that originates from the reversible redox chemical reactions on electrode surfaces. We further summarize the correlation of pseudocapacitance with the current densities and sweep rates for the porous MoN, TiN and $Ta_5N_6$ single crystals as shown in Supplementary Fig. 18. The linear fitting gives the correlation coefficients of 0.954–0.999, which indicates that the capacitance performance is mainly dominated by the pseudocapacitance from the redox reactions on electrode surfaces. As shown in Fig. 5f, we summarize the areal capacitance with metallic porous MoN single crystals as a function of sweeping current densities. The porous MoN single crystal reaches a very high value of 8.8 F cm$^{-2}$ at a current density of 5 mA cm$^{-2}$ in a CD measurement, and remains 3 F cm$^{-2}$ even at 100 mA cm$^{-2}$, indicating the excellent performance with metallic porous single crystals at high constant current charge-discharge process. We further summarize the reported areal pseudocapacitance with different nano-scale materials and show that the areal pseudocapacitance with porous MoN single crystals is among the first level with sulfide electrode and ~3–10 times higher than reported values with most oxide and nitride electrodes as shown in Supplementary Table 3[38–50].

We further grow porous MoN single crystals with the thickness of 18 and 36 μm and measure the specific area capacitance normalized relative to their BET surface areas. We get a very high normalized area capacitance value of more than 3500 μF cm$^{-2}$

which is among the highest values of current materials that exhibit pseudocapacitance and hundreds to thousands times larger than the value of the EDLC of carbon materials[51]. The pseudocapacitance is mainly dominated by the surface chemisorption on nitride surfaces. We further grow the porous MoN single crystals with the thickness ranging to 500 μm and investigate the correlation of pseudocapacitance with the electrode thickness. As shown in Supplementary Fig. 19, it is observed that the effective chemisorption takes place below the thickness of ~70 μm in the porous microstructures which delivers the pseudocapacitance for porous single crystals. The highest specific area capacitance is therefore observed for the porous MoN single crystals with the thickness of 70 μm. We further summarize the relationship between coordination structure, surface area, gravimetric capacitance and areal capacitance for porous MoN, TiN, and $Ta_5N_6$ single crystals as shown in Supplementary Fig. 20. Coordinatively unsaturated metal-nitrogen sites account for the chemisorption while reasonable adsorption energy gives rise to enhanced pseudocapacitance. The porous MoN single crystal shows the highest gravimetric capacitance of ~2000 F g$^{-1}$ in alkaline electrolyte and ~800 F g$^{-1}$ in acidic electrolyte. And the area capacitance of ~8 F cm$^{-2}$ in alkaline electrolyte and ~5 F cm$^{-2}$ in acidic electrolyte are also obtained with porous MoN single crystal.

## Discussion
In conclusion, we demonstrate the long-range ordering of the active site on metallic porous nitride single crystals to enhance pseudocapacitance. The synergy of porous microstructure, electronic conduction and ordered active sites in metallic porous single-crystalline electrode facilitates the fast reversible redox reactions and therefore contributes to the enhanced

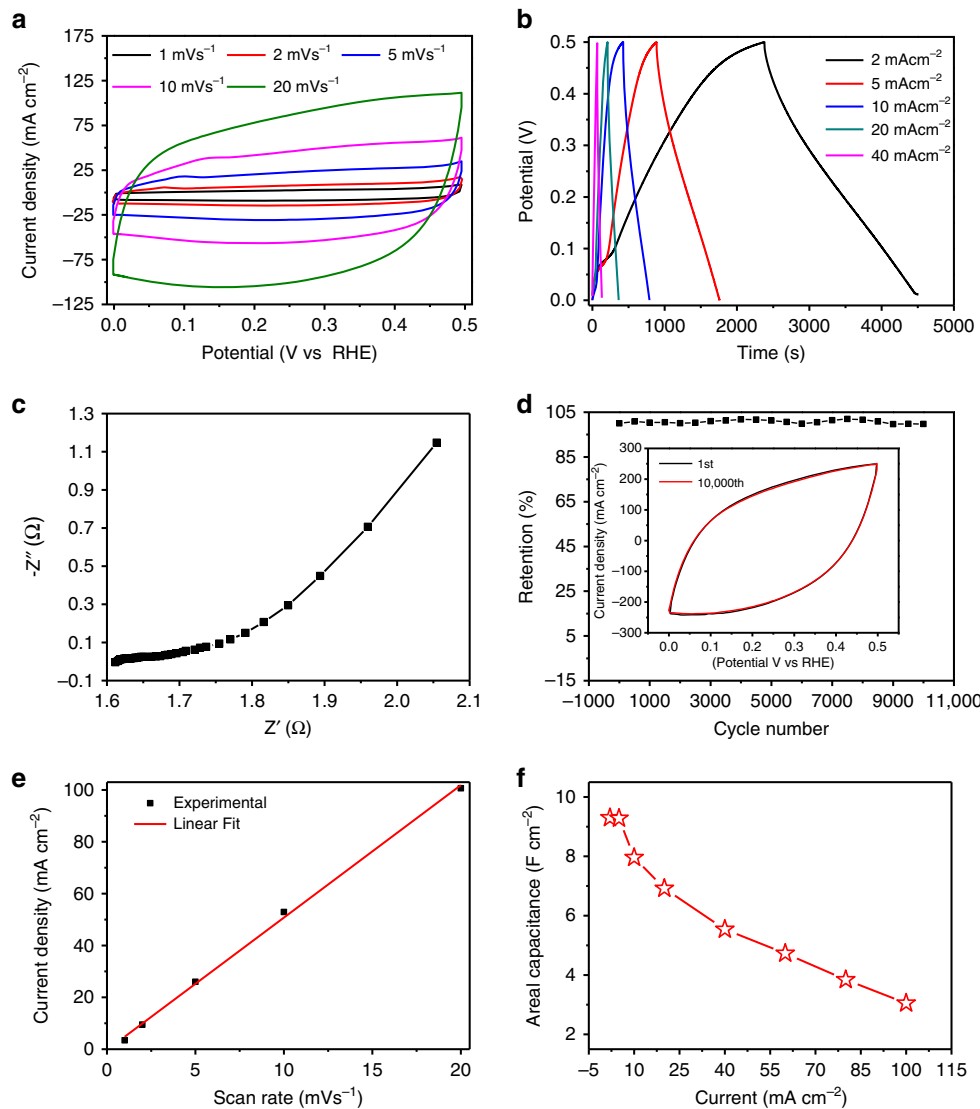

**Fig. 5** Electrochemical performance of MoN single crystal in 1 M KOH electrolyte. **a** CV curves at different scan rate of porous MoN single crystal. **b** GCD curves at different current densities of the MoN crystal. **c** Nyquist plots of MoN crystal electrode before cycling. **d** Cycling performance of MoN single crystal at a scan rate of 50 mV s⁻¹, inset shows the CV curves of the 1st and 10,000th cycles. **e** Linear dependence of current density on scan rate of CVs at scan rates from 1 to 20 mV s⁻¹. **f** Areal capacitance as a function of current densities of MoN single crystal

pseudocapacitance. We show the improved gravimetric and areal pseudocapacitance and excellent cycling stability with porous metal-nitride single crystals both in acidic and alkaline electrolytes. The current work would shed a light on the growth of metallic porous single crystals to tailor resolved local structures with unsaturated coordination on surface and hence to engineer the fast redox reactions in chemisorption that results in pseudocapacitance. The lattice reconstruction strategy would also open a new route to grow porous single crystals in wealth of other materials to enhance the capability of the materials themselves.

## Methods

**Experiment**. The MoN single crystal is prepared by directly treating PbMoO₄ single crystals in ammonia atmosphere using chemical vapor deposition system[7–9]. The PbMoO₄ substrates (20 × 10 × 0.5 mm), both side polished, are washed with ultrasonic wave in distilled water and organic solvents (ethanol, acetone) for several times. The substrates are dried in a stream of dry nitrogen and immediately put into an alumina tube furnace. The substrates are annealed in NH₃ at 750 °C for 10 h under the pressure of 25–500 Torr. Ammonia gas (99.999%) is passed through the tube at a flow rate of 600 mL min⁻¹. Then the outlet gas is cooled with cycling

water to room temperature while the Pb and PbO are collected in the steel chamber in the form of solid powder out of the furnace. The solid Pb/PbO powders are collected and disposed in the form of toxic chemicals. We show the optical photo of the experimental setup in Supplementary Fig. 21.

We use 100 mL of 1 M KOH and 0.5 M H₂SO₄ as the electrolyte solutions. The electrochemical experiments are carried out using a three electrode setup. A Pt foil with a dimension of 1 cm × 1 cm works as the counter electrode and a saturated calomel electrode (0.244 V versus SHE) is used as reference electrode. All experiments are performed using an electrochemical workstation (IM6 Zahner, Germany). The electrochemical impedance spectra (EIS) are recorded in the frequency range of 0.1–10⁵ Hz at the applied amplitude of AC potential of 5 mV. The Nyquist plots are recorded from the fitted EIS results. Conductivities of the synthesized electrode are measured by the four point probe method in PPMS (Quantum Design).

**Calculation**. Density functional theory (DFT) calculations are performed using Vienna Ab Initio Simulation Package (VASP) code[52]. Based on the projector augmented wave (PAW) approach, the plane-wave cutoff energy of 500 eV is used, which gives well converged relative energies for the system. The Perdew-Burke-Ernzerhof (PBE) function is chosen here to describe the exchange correlation interactions. The energies and residual force are converged to 10⁻⁶ eV with an d 0.02 eV Å⁻¹ in the process of electronic and geometric optimizations. We calculate the unit cell and obtain the lattice parameters of MoN: a = b = 5.73 Å and c = 5.61 Å (a 6 × 6 × 6 k-point grid). A p(2 × 2) superstructure with Mo termination of the

[001] surface is used for the slab model[34,53]. In this model, which includes three layers, the atoms in the bottom layers are frozen to its bulk geometry during the optimization and other atoms are fully relaxed. A 24-Å vacuum region is insert along the $c$ direction to avoid the interactions between the slab and its repeating image. A $3 \times 3 \times 1$ k-point grid is used for Brillouin zone sampling. Spin-polarization is considered for all calculation.

## Data availability

The data that support the findings of this study are available from the corresponding author upon request. All reported data are included in the paper and supplementary materials. The Source Data can be downloaded from: https://yunpan.360.cn/surl_yuZg3ueVvAY (Code: 5f09).

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

## Acknowledgements

We acknowledge the funding support from the National Key Research and Development Program of China (2017YFA0700102), Natural Science Foundation of China (91845202), Dalian National Laboratory for Clean Energy (DNL180404), and Strategic Priority Research Program of Chinese Academy of Sciences (XDB2000000).

## Author contributions

S.X. conducted the work related to metallic porous MoN and Ta5N6 single crystals. G.L. conducted the work related to metallic porous TiN single crystal. G.L. and L.J. helped to analyze the data and draw the figures. H.L. conducted the DFT calculation. K.X. supervised the work.

## Competing interests

The authors declare no competing interests.
