## [Peer Review File · Nature Communications]

Reviewers' comments:

Reviewer #1 (Remarks to the Author):

Authors reported higher areal pseudocapacitance, high rate capability and excellent cycling stability both in acidic and alkaline electrolyte with porous MoN, Ta₅N₆ and TiN single crystals. They predicted that long-range Ordering of Active Sites at 2 cm Scale is the main reason for higher Pseudocapacitance. Their work is of potential interest. The manuscripts is well written. I feel the manuscript needs a MAJOR REVISION on the following points before accepting for publication

1. Author mentioned (in line 100), 'the higher adsorption energy of OH than H on MoN surface would deliver higher pseudocapacitance'- Authors should compare reported adsorption energy of H and OH on the similar surfaces. While calculating adsorption energy, whether authors considered ion H⁺ or only H atom
2. While computing adsorption energy, did author take dispersion correction ? if yes which scheme they have used?
3. Authors should compare computed lattice parameters and band gap (if any) of the systems MoN, Ta₅N₆ and TiN with reported experimental and theoretical value?
4. Text in Fig. 1a not visible; Fig 3C XPS (inset) not visible
5. Author should provide proper explanation that why MoN exhibits higher pseudocapacitances compared to other two nitride compounds
6. In the introduction, Authors should refer supercapacitance work on other transition metal based systems { J. Phys. Chem. C 2017, 121, 18992–19001; ACS Appl. Mater. Interfaces 2017, 9, 9640–9653; J. Phys. Chem. C 2018, 122, 21140–21150}
7. Author should provide a table comparing areal Pseudocapacitance for similar material reported in the literature

Reviewer #2 (Remarks to the Author):

This paper reports several metallic porous nitrides with single crystal structures that showed very high areal pseudocapacitances. Although the microstructure of these materials is interesting, the concept of preparation is not new (Ref. 4-6). More importantly, the study on their electrochemical properties was too routine, and the real charge storage mechanism did not clearly analyzed. From a fundamental point of view, it is hard to see highly important information coming from this study, unfortunately. In addition, some issues should be addressed for reinforcing the manuscript. Therefore, it is unsuitable for publication in Nature Communications.

1. In introduction section, the authors said that "Pseudocapacitance originates from reversible redox reactions on electrode surface regions". However, pseudocapacitance can be mainly divided into redox pseudocapacitance and insertion pseudocapacitance. The insertion pseudocapacitance refers to the insertion of alkali metal ions or protons into the host materials during electrochemical sweeping. Therefore, the authors should explain pseudocapacitance more accurately.
2. The authors state that "In this work, we demonstrate a disruptive approach of lattice reconstruction strategy to create long-range ordering of the active site on metallic porous nitride single crystals". Specifically, they grew metallic porous nitrides single crystals through a lattice reconstruction strategy in vacuum ammonia atmosphere at high temperature. However, this method has been reported before (Ref 4-6), especially porous TiN. Therefore, the relative contents might be not the first demonstration in this manuscript.
3. During the preparation of porous MoN, the evaporation of Pb/O from lattice is necessary. As we know, Pb is highly toxic element, and its evaporation is not conducive to the environment. Therefore, such preparation method of electrode materials is not promising for applications of electrochemical devices.
4. How about the content of residual oxygen in final materials?

5. The structural feature of unsaturated metal-nitrogen active sites in final nitrides should be clearly analyzed. Moreover, the effect of such structure on the electrochemical properties should be clearly explained and verified.
6. The nearly rectangular CV curves and the triangular GCD curves indicated the good capacitance property of MoN. In certain extent, such behavior looks like the EDLC of carbon materials. Thus, it is strongly recommended to study the real charge storage mechanism in depth.
7. The authors state that "The metallic porous MoN single crystal reaches a very high value of 8.8 F cm⁻² at current density of 5 mA cm⁻² in a CD measurement, which represents the highest pseudocapacitance and is ~3-10 times higher than reported values with oxide and nitride electrodes". I found that the main reason is that they used the PbMoO₄ substrates (20 mm × 10 mm × 0.5 mm). That to say, the very thick electrode (might be about 500 μm) resulted in the high areal capacitance shown in this paper, instead of the intrinsic characteristic MoN. So, the comparison listed in the Figure 5f is not fully fair competition.
8. I cannot find the solid data to support the excellent rate performance the authors mentioned in this paper.
9. What is the deep reason that porous MoN showed excellent cycling stability?

Reviewer #3 (Remarks to the Author):

In this work, the authors developed a series porous metal nitride as electrode for SCs. The obtained material showed attractive performance when applied for SCs. However, this paper do not provide some remarkable scientific viewpoints. In some way, the characterization supported points in this work. I recommend this paper for publication in Nature Communications after major revision.

1. The authors claimed they prepared metal nitride single crystals such as MoN, Ta₅N₆ and TiN, however, the normal XRD test can't confirm the single crystal quality, the FWHM and X-ray rocking curve should be given.
2. In this manuscript, the authors consider the unsaturated metal-nitrogen active sites benefits for the pseudocapacitance, those active sites in what way absorb the electrolyte ions, what kinds of redox reaction takes place? Some characterization to prove the unsaturated metal-nitrogen should be given, like EPR.
3. The author claimed microstructures of porous MoN in the paper, how to define? As an important characterization method, the specific surface area and pore size distribution should be given.
4. The authors think they obtain the highest areal pseudocapacitance of 8.8 F cm⁻² for the MoN electrode, however, I suggest they should compare more recent articles when they say the highest capacitance, such as the Ni₃S₂ can achieve 21.54/9.21 F cm⁻² at 2/5 mA cm⁻² (J. Mater. Chem. A, 2018, 6, 22115); the NiCo₂S₄ obtain the areal capacitance of 14.39 F cm⁻² at 5 mA cm⁻² (J. Power Sources, 2014, 254, 249); and the capacitance of the MnO₂ is 44.13 F cm⁻². (Joule, 2019, 3, 459)
5. The conductivity of the MoN, TiN and Ta₅N₆ single crystals are 0.77 × 10⁴, 0.45 × 10⁴ and 0.8 × 10⁴ S cm⁻¹, how did you get these data, references or measurement?
6. The adsorption energy from the theoretical calculation is inappropriate, from the text, the adsorption energy of H and OH on (001) Ta₅N₆ surface are -1.10~-1.32 eV and -2.19~-2.56 eV, respectively, so the higher adsorption energy should be OH, however, it is not consistent with the experimental result.
7. The MoN and TiN surface deliver higher pseudocapacitance in alkaline electrolyte, and why the Ta₅N₆ obtain the higher capacitance in acidic electrolyte? Please give a reasonable explanation?
8. The voltage windows of CV and GCD tests are different (Figure 5a, 5b and 5d), how does the author choose the right voltage range?
9. In Figure 5c EIS, the unit on the Y-axis should be -Z'', the sentence "the high-frequency region indicates very fast species diffusion in MoN single crystal" is a conceptual mistake.

Response to reviewers

Reviewers' comments:

Reviewer #1 (Remarks to the Author):

Authors reported higher areal pseudocapacitance, high rate capability and excellent cycling stability both in acidic and alkaline electrolyte with porous MoN, Ta₅N₆ and TiN single crystals. They predicted that long-range Ordering of Active Sites at 2 cm Scale is the main reason for higher Pseudocapacitance. Their work is of potential interest. The manuscripts is well written. I feel the manuscript needs a MAJOR REVISION on the following points before accepting for publication.

Answer: Thank you very much for your comments. We have fully conducted supplementary experiments and substantially revised the manuscript. We sincerely wish these revision would well fit your requirements.

1. Author mentioned (in line 100), ‘the higher adsorption energy of OH than H on MoN surface would deliver higher pseudocapacitance’- Authors should compare reported adsorption energy of H and OH on the similar surfaces. While calculating adsorption energy, whether authors considered ion H⁺ or only H atom.

Answer: Thank you very much for your comments. In our work, we consider the H⁺ and OH⁻ as the adsorbed species when we calculate the adsorption energy on the crystal surfaces. The adsorption energy of H and OH on similar surfaces have been reported and we further summarize them as listed in **Supplementary Table 2**. The adsorption energy is normally below 0 which indicates an exothermic process that favors the chemisorption on different surfaces of metallic substrates. The calculated adsorption energy in our work is well consistent with the reported values in references.

Surface	Energy (eV)	Surface	Energy (eV)
Ta ₃ N ₅ (100)-OH ^[1]	-2.3 ~ -3.3	MoC(111)-H ^[4]	-2.90 ~ -3.26
Ta(001)-H ^[2]	0.70 ~ -0.44	Mo(110)-OH ^[5]	-0.61
Ta(110)-H ^[3]	-2.35 ~ -3.27	Mo ₂ N(100)-H ^[6]	-2.74 ~ -3.14
Ta(100)-H ^[3]	-2.31 ~ -2.93	TiN-2(OH) ^[7]	-1.18 ~ -7.66
Mo(110)-H ^[3]	-1.56 ~ -3.08	TiO ₂ (001)-H ^[8]	-0.13 ~ -1.56
Mo(100)-H ^[3]	-1.89 ~ -2.99	TiO ₂ (100)-H ^[8]	-0.03 ~ -2.68

Supplementary Table 2. The adsorption energies of H and OH on different surfaces. The data are collected from references and summarized as listed in the table.

References:

[1] Watanabe, E., Ushiyama, H. & Yamashita, K. Theoretical studies on the mechanism of oxygen reduction reaction on clean and O-substituted Ta₃N₅(100) surfaces. *Catal. Sci. Technol.* **5**, 2769-2776 (2015).

- [2] Xu, L., Xiao, H. Y. & Zu, X. T. First-principles study on the geometry and stability of hydrogen on the Ta(001) (1×1) surface. *Surf. Rev. Lett.* **12**, 809-817 (2005).
- [3] Ferrin, P., Kandoi, S., Nilekar, A. U. & Mavrikakis, M. Hydrogen adsorption, absorption and diffusion on and in transition metal surfaces: A DFT study. *Surf. Sci.* **606**, 679-689 (2012).
- [4] Wang, T., Li, Y.W., Wang, J., Beller, M. & Jiao, H. Dissociative Hydrogen Adsorption on the Hexagonal Mo₂C Phase at High Coverage. *J. Phys. Chem. C* **118**, 8079-8089 (2014).
- [5] Norskov, J. K. *et al.* Trends in the exchange current for hydrogen evolution. *J. Electrochem. Soc.* **152**, J23-J26 (2005).
- [6] Zhao, J. *et al.* Insights into the Mechanism of Ammonia Decomposition on Molybdenum Nitrides Based on DFT Studies. *J. Phys. Chem. C* **123**, 554-564 (2019).
- [7] Seifitokaldani, A., Savadogo, O. & Perrier, M. Density Functional Theory (DFT) Computation of the Oxygen Reduction Reaction (ORR) on Titanium Nitride (TiN) Surface. *Electrochim. Acta* **141**, 25-32 (2014).
- [8] Calatayud, M. & Minot, C. Effect of relaxation on structure and reactivity of anatase (100) and (001) surfaces. *Surf. Sci.* **552**, 169-179 (2004).

2. While computing adsorption energy, did author take dispersion correction? if yes which scheme they have used?

Answer: Thank you very much for your comments. The dispersion correction is not adopted during the calculation in our work. The calculated adsorption energies generally well fit the experimental phenomenon in our work.

3. Authors should compare computed lattice parameters and band gap (if any) of the systems MoN, Ta₅N₆ and TiN with reported experimental and theoretical value?

Answer: Thank you very much for your comments. We have compared the calculated parameters with theoretical and experimental values as listed in **Supplementary Table 1**. The computed values of MoN, TiN and Ta₅N₆ are well consistent with the theoretical and experimental values, which further indicates the reliability of theoretical calculation in our work.

We further conduct the calculation of the band gaps and electronic structures of the MoN, TiN and Ta₅N₆. These nitrides are typical electronic conductors with band gaps of 0 eV as shown in **Supplementary Figure 9**, which further validates the metallic properties of these porous nitride single crystals. We can clearly see from the DOS map that the valence band is mainly composed of 2s and 2p orbital of N atoms and 3d orbital transition metal atoms. While the conduction band is dominated by the 3d orbital of transition metal atoms and the 2p orbital of N atoms has a small contribution.

	Space group	Computed values (Å)	Theoretical Value (Å)	Experiment (Å)
MoN	P63/MMC	a=b=5.73, c=5.61	a=5.71, c=5.63 ^[1]	a=5.72 c=5.60 ^[2]
TiN	FM-3M	a=b=c=4.244	a=b=c=4.254 ^[3]	a=b=c=4.240 ^[4]
Ta ₅ N ₆	P63/MMC	a=b=5.176, c=10.35	a=b=5.22, c=10.46 ^[5]	a=b=5.176 c=10.35 ^[6]

Supplementary Table 1. The comparison of calculated parameters, theoretical parameters and experimental values of of MoN, TiN and Ta₅N₆ while the theoretical and experimental values are obtained from references as listed below.

References:

- [1] Kanoun, M. B., Goumri-Said, S. & Jaouen, M. Structure and mechanical stability of molybdenum nitrides: A first-principles study. *Phys. Rev. B* **76**, 134109 (2007).
- [2] Ihara, H., Kimura, Y., Senzaki, K., Kezuka, H. & Hirabayashi, M. Electronic structures of B1 MoN, fcc Mo₂N, and hexagonal MoN. *Phys. Rev. B* **31**, 3177-3178 (1985).
- [3] Moreno, J. J. G. & Nolan, M. Ab Initio Study of the Atomic Level Structure of the Rutile TiO₂(110) -Titanium Nitride (TiN) Interface. *ACS Appl. Mater. Interfaces* **9**, 38089-38100 (2017).
- [4] Hoerling, A. *et al.* Thermal stability, microstructure and mechanical properties of Ti_{1-x}Zr_xN thin films. *Thin Solid Films* **516**, 6421-6431 (2008).
- [5] Wang, J., Jiang, J., Chen, J., Li, Y. & Ma, A. Theoretical study of the oxygen impurity doped Ta₅N₆. *Comput. Mater. Sci.* **143**, 368-373 (2018).
- [6] Petrunin, V.E., Sorokin, N.I., Borovinskaya, I.P. & Pityulin, A.N. Stability of cubic tantalum nitrides during heat treatment. *Powder Metall. Met. Ceram.* **19**, 191-192 (1980).

Supplementary Figure 9. The band gaps and density of states of the three different nitrides. (a-c) The band gaps and electronic structures of MoN. (d-f) The band gaps and electronic structures of Ta₅N₆. (g-i) The band gaps and electronic structures of TiN.

4. Text in Fig. 1a not visible; Fig 3C XPS (inset) not visible

Answer: Thank you very much for your comments. We have made it different in revision.

5. Author should provide proper explanation that why MoN exhibits higher pseudocapacitances compared to other two nitride compounds.

Answer: Thank you very much for your comments and we have provided the corresponding explanation in the parts of results and discussion. Metallic porous nitride single crystals have the advantages of large surface area, porous microstructure, high conductivity and ordered active sites with unsaturated coordination. The structural coherence in single crystals would confine the

ordered active sites with unsaturated coordination in lattice, which therefore creates the electrochemically active surface to host fast redox reactions in chemisorption and leads to enhanced pseudocapacitance. And the continuously twisted surfaces in the microstructures also produce surface stress that further favors the formation of high energy surfaces.

We further conduct supplementary experiments to test the specific surface area of the three different porous single crystals in revision. As shown in **Supplementary Figure 6**, the specific surface area of porous MoN single crystal is $14.1 \text{ m}^2 \text{ g}^{-1}$, which is about 7 and 3 times higher than 2.1 and $5.9 \text{ m}^2 \text{ g}^{-1}$ for the porous Ta_5N_6 and TiN single crystals, respectively. Although the three nitrides are all in single-crystalline states, the difference in specific surface areas may influence the pseudocapacitances. We believe that the larger surface areas of porous MoN single crystal would deliver higher pseudocapacitances in contrast to the other two nitride single crystals. On the other hand, the redox activity of Mo is much higher than Ti but comparable to Ta element, which may also help the formation of performance difference between the three different porous single crystals at current status in our work.

Supplementary Figure 6. The specific surface area and average pore size of the three porous nitride single crystals. **a** Specific area of MoN, Ta_5N_6 and TiN. **b** average pore size of porous MoN, Ta_5N_6 and TiN single crystals. The error bars represent standard deviation in repeated measurements.

6. In the introduction, Authors should refer supercapacitance work on other transition metal based systems {J. Phys. Chem. C 2017, 121, 18992–19001; ACS Appl. Mater. Interfaces 2017, 9, 9640–9653; J. Phys. Chem. C 2018, 122, 21140–21150}.

Answer: Thank you very much for your suggestions. Yes, we have cited these references in revision to reinforce the background of supercapacitors in introduction part.

7. Author should provide a table comparing areal Pseudocapacitance for similar material reported in the literature

Answer: Thank you very much for your comments and we have provided the comparison in revision. The comparison between our porous nitride single crystals and reported materials is shown in **Supplementary Table 3**. The areal capacitance of porous MoN single crystals is 8.8 F cm^{-2} at 5 mA cm^{-2} . And it should be noted that the structural coherence of porous single crystals delivers remarkable stability with the retention of 100% after 10000 cycles, which has demonstrated obvious advantages in contrast to most of the reported materials.

Supplementary Table 3. The areal capacitance, stability and voltage window of reported materials. The data is collected from the references as listed below.

Materials	Areal capacitance	Current load or scan rate	Retention	Voltage window	Reference
Porous MoN single crystal	8.8 Fcm ⁻²	5 mAcm ⁻²	100% after 10000 cycles	0-0.5 V vs Ag/AgCl	This work
Nb ₄ N ₅ @NC	0.226 Fcm ⁻²	0.5 mVs ⁻¹	100% after 2000 cycles	0-1.0 V vs Ag/AgCl	1
MnO ₂ nanorods	0.22 Fcm ⁻²	0.75 mAcm ⁻²	95.5% after 5000 cycles	0-1.8 V vs SCE	2
HG-Ti ₃ C ₂	4 Fcm ⁻²	5 mVs ⁻¹	90% after 10000cycles	-1.1-0.2 V vs SCE	3
Co ₃ O ₄ @MnO ₂ nanowire array	0.7 Fcm ⁻²	4 mAcm ⁻²	97.3% after 5000 cycles	-0.2-0.6 V vs Ag/AgCl	4
Mesoporous NiCo ₂ O ₄ nanosheets	3.51 Fcm ⁻²	1.8 mAcm ⁻²	93.3% after 3000 cycles	-0.2-0.6 V vs SCE	5
Carbon fiber cloth /MnO ₂ /CNTs	3.416 Fcm ⁻²	2 mVs ⁻¹	100% after 1500 cycles	0-1.0 V	6
WO _{3-x} /MoO _{3-x} Core/Shell nanowires	0.216 Fcm ⁻²	2 mAcm ⁻²	75% after 10000 cycles	0-1.9 V vs RHE	7
Activated carbon fiber	1.56 Fcm ⁻²	5 mAcm ⁻²	100% 20000 cycles	0-1.0 V vs SCE	8
Ti-Doped Fe ₂ O ₃ @PEDOT	1.15 Fcm ⁻² □□	1 mAcm ⁻²	85.4% after 600 cycles	0-1.6 V vs SCE	9
PANI/Au/paper	0.8 Fcm ⁻²	1 mAcm ⁻²	100% after 10000 cycles	0-0.8 V vs SCE	10
Hierarchical urchin-like Ni ₃ S ₂	21.54 Fcm ⁻²	2 mAcm ⁻²	59.2% after 1000 cycles	0-0.6 V vs SCE	11
NiCo ₂ S ₄ nanotube arrays on Ni foam	14.39 Fcm ⁻²	5 mAcm ⁻²	92% after 5000 cycles	0-0.6 vs Hg/HgO	12
4-mm-thick Ultrahigh Loading MnO ₂	44.13 Fcm ⁻²	0.5 mAcm ⁻²	92.9% after 20000cycles	0-0.5 V vs SCE	13

References:

- [1] Cui, H. *et al.* Niobium Nitride Nb₄N₅ as a New High-Performance Electrode Material for Supercapacitors. *Adv. Sci.* **2**, 1500126 (2015).
- [2] Zhai, T. *et al.* Oxygen vacancies enhancing capacitive properties of MnO₂ nanorods for wearable asymmetric supercapacitors. *Nano Energy* **8**, 255-263 (2014).
- [3] Lukatskaya, M. R. *et al.* Ultra-high-rate pseudocapacitive energy storage in two-dimensional transition metal carbides. *Nat. Energy* **2**, 17105 (2017).
- [4] Liu, J. *et al.* Co₃O₄ Nanowire@MnO₂ Ultrathin Nanosheet Core/Shell Arrays: A New Class of High-Performance Pseudocapacitive Materials. *Adv. Mater.* **23**, 2076-2081 (2011).
- [5] Zhang, G. & Lou, X. W. General Solution Growth of Mesoporous NiCo₂O₄ Nanosheets on Various Conductive Substrates as High-Performance Electrodes for Supercapacitors. *Adv. Mater.* **25**, 976-979 (2013).
- [6] Sumboja, A., Foo, C. Y., Wang, X. & Lee, P. S. Large Areal Mass, Flexible and Free-Standing Reduced Graphene Oxide/Manganese Dioxide Paper for Asymmetric Supercapacitor Device. *Adv. Mater.* **25**, 2809-2815 (2013).
- [7] Xiao, X. *et al.* WO_{3-x}/MoO_{3-x} Core/Shell Nanowires on Carbon Fabric as an Anode for All-Solid-State Asymmetric Supercapacitors. *Adv. Energy Mater.* **2**, 1328-1332 (2012).
- [8] Dong, L. *et al.* Simultaneous Production of High-Performance Flexible Textile Electrodes and Fiber Electrodes for Wearable Energy Storage. *Adv. Mater.* **28**, 1675-1681 (2016).
- [9] Zeng, Y. *et al.* Advanced Ti-Doped Fe₂O₃@PEDOT Core/Shell Anode for High-Energy Asymmetric Supercapacitors. *Adv. Energy Mater.* **5**, 1402176 (2015).
- [10] Yuan, L. *et al.* Paper-Based Supercapacitors for Self-Powered Nanosystems. *Angew. Chem. Int. Ed.* **51**,

4934-4938 (2012).

[11] He, Q., Wang, Y., Liu, X. X., Blackwood, D. J. & Chen, J. S. One-pot synthesis of self-supported hierarchical urchin-like Ni_3S_2 with ultrahigh areal pseudocapacitance. *J. Mater. Chem. A* **6**, 22115-22122 (2018).

[12] Chen, H. *et al.* In situ growth of NiCo_2S_4 nanotube arrays on Ni foam for supercapacitors: Maximizing utilization efficiency at high mass loading to achieve ultrahigh areal pseudocapacitance. *J. Power Sources* **254**, 249-257 (2014).

[13] Yao, B. *et al.* Efficient 3D Printed Pseudocapacitive Electrodes with Ultrahigh MnO_2 Loading. *Joule* **3**, 1-12 (2019).

Reviewer #2 (Remarks to the Author):

This paper reports several metallic porous nitrides with single crystal structures that showed very high areal pseudocapacitances. Although the microstructure of these materials is interesting, the concept of preparation is not new (Ref. 4-6). More importantly, the study on their electrochemical properties was too routine, and the real charge storage mechanism did not clearly analyzed. From a fundamental point of view, it is hard to see highly important information coming from this study, unfortunately. In addition, some issues should be addressed for reinforcing the manuscript. Therefore, it is unsuitable for publication in Nature Communications.

Answer: Thank you very much for your comments. We have fully and carefully conducted supplementary experiments and substantially revised the manuscript. We grow metallic porous nitride single crystals at 2 cm scale to combine the advantages of large surface area with porous microstructure, high conductivity with metallic states and ordered active sites with unsaturated coordination. We therefore show the enhanced areal pseudocapacitance and cycling stability with porous single crystals. The long-range ordering of resolved local surface defect structures of transition metals with unsaturated coordination account for the fast redox reactions in chemisorption while the high conductivity together with porous microstructure facilitate the charge transfer and species diffusion in electrodes.

1. In introduction section, the authors said that “Pseudocapacitance originates from reversible redox reactions on electrode surface regions”. However, pseudocapacitance can be mainly divided into redox pseudocapacitance and insertion pseudocapacitance. The insertion pseudocapacitance refers to the insertion of alkali metal ions or protons into the host materials during electrochemical sweeping. Therefore, the authors should explain pseudocapacitance more accurately.

Answer: Thank you very much for your comments. Yes, we fully agree with the reviewer on the definition of pseudocapacitance and we have corrected it in revision. Pseudocapacitance originates from reversible redox reactions on electrode surface regions or insertion of alkali metal ions or protons into the host materials during electrochemical sweeping. In our work, the pseudocapacitance actually originates from the reversible redox reactions on surface regions of nitride single-crystalline electrodes.

Supplementary Figure 18. The linear fitting of dependence of current density on scan rate of CVs at 0.35V. **a** MoN in KOH. **b** MoN in H₂SO₄. **c** Ta₅N₆ in KOH. **d** Ta₅N₆ in H₂SO₄. **e** TiN in KOH. **f** TiN in H₂SO₄.

The linear dependence of capacitance performance on the current densities and sweep rates clearly confirms the typical characteristics of pseudocapacitance that originates from the reversible redox chemical reactions on electrode surfaces as shown in Figure 5e, and Supplementary Figure 6e, 7e, 8e, 9e and 10e. We further summarize the correlation of pseudocapacitance with the current densities and sweep rates for the porous MoN, TiN and Ta₅N₆ single crystals as shown in Supplementary Figure 18. The linear fitting gives the correlation coefficients of 0.954-0.999, which indicates that the capacitance performance is mainly dominated by the pseudocapacitance from the redox reactions on electrode surfaces.

2. The authors state that “In this work, we demonstrate a disruptive approach of lattice reconstruction strategy to create long-range ordering of the active site on metallic porous nitride single crystals”. Specifically, they grew metallic porous nitrides single crystals through a lattice reconstruction strategy in vacuumed ammonia atmosphere at high temperature. However, this method has been reported before (Ref 4-6), especially porous TiN. Therefore, the relative contents

might be not the first demonstration in this manuscript.

Answer: Thank you very much for your comments. Yes, the growth of porous single crystals has been reported in our previous work. We have carefully revised our manuscript according to your comments. Here we focus on the long-range ordering of active sites with unsaturated nitrogen coordination that give rise to electrochemically active surfaces. The combination of large surface area with porous microstructure, high conductivity with metallic states and ordered active sites with unsaturated coordination delivers enhanced areal pseudocapacitance while the structural coherence significantly enhances the cycling stability of nitride electrodes.

3. During the preparation of porous MoN, the evaporation of Pb/O from lattice is necessary. As we know, Pb is highly toxic element, and its evaporation is not conducive to the environment. Therefore, such preparation method of electrode materials is not promising for applications of electrochemical devices.

Answer: Thank you very much for your comments. It is reasonable to say that the Pb element is toxic and it would be good if the experiment can be environmentally friendly. Here we demonstrate the growth of porous MoN single crystal from the parent PbMoO₄ single crystal through a lattice reconstruction strategy in a vacuum system at ~750 °C as shown in **Supplementary Figure 21**. In this process, the Pb and O elements are removed from the lattice of PbMoO₃ accompanied by the substitution of O with N during the nitridation to grow the large size porous nitride single crystals.

A : Gas inlet; B: Heater; C: Alumina ceramic tube; D: Circulating water; E: Gas outlet; F: Pressure controller; G: Gas flow controller; H: Heating controller I: Sample chamber; J: Steel cooling chamber

Supplementary Figure 21. The digital photo of our vacuum system which consists of gas inlet, gas outlet, cycling water, alumina tube heating system and controlling systems. **a** Vacuum system. **b** Control panel. **c** The chamber for preparing sample. **d** Pushing samples into the chamber. **e** Sealing the chamber for preparing the sample. **f** The outlet tip of the system. **g** The steel cooling chamber of the vacuum system.

We use ammonia together with nitrogen as reaction gas to remove Pb and O elements which are pumped to gaseous phases in the vacuum system at ~750 °C. Then the outlet gas is cooled with cycling water to room temperature while the Pb and PbO are collected in the steel chamber in the form of solid powder out of the furnace. The solid Pb/PbO powders are disposed according to the

regulation managements of toxic chemicals in our institute. Actually, the environmental friendly parent single crystals will be considered for the growth of porous single crystals using the approach of lattice reconstruction strategy in our future work.

4. How about the content of residual oxygen in final materials?

Answer: Thank you very much for your comments and we further conduct supplementary experiments. No oxygen residual is observed in the porous MoN single crystals according our element analysis as shown in the EDX in **Supplementary Figure 2**. The parent crystals are completely nitrified into MoN single crystal after sufficient nitridation treatment in our work.

Supplementary Figure 2. The element analysis of MoN single crystal. No oxygen residual is observed. The mole ratio between N and Mo is approximately at 1.

5. The structural feature of unsaturated metal-nitrogen active sites in final nitrides should be clearly analyzed. Moreover, the effect of such structure on the electrochemical properties should be clearly explained and verified.

Answer: Thank you very much for your comments. We use high-resolution Cs-corrected STEM to analyze the coordination structures of the three porous single crystals. We use high-sensitive low energy ion scattering (HS-LEIS) with He⁺ (3 keV) and Ne⁺ (5 keV) ions sources to detect the outmost atomic surface state of porous MoN single crystals. We summarize the surface characteristics of the three porous single crystals as shown in **Supplementary Figure 8**.

The porous MoN, TiN and Ta₅N₆ single crystals atomically terminate with Mo, Ti and Ta atom layer, respectively, with nitrogen atom layer stacking beneath the surface metal atom layer. The ordered active sites are confirmed to be Mo-N_{1/2}, Ti-N_{5/6} and Ta-N_{3/5} which give rise to electrochemically active surfaces to host fast redox reactions in chemisorption. According to the charge balance, only pseudocapacitance with OH⁻ adsorption is observed for porous TiN single crystal while pseudocapacitance both with OH⁻ and H⁺ adsorption is observed for porous MoN and Ta₅N₆ single crystals. Our calculation further validates the larger energies of chemisorption of OH⁻ and H⁺ on MoN and Ta₅N₆ surfaces. The TiN surfaces favor the chemisorption of OH⁻ but limits the chemisorption of H⁺ in electrolyte.

Supplementary Figure 8. (a,c,e) The high-sensitive low energy ion scattering (HS-LEIS) with He⁺ (3 keV) and Ne⁺ (5 keV) ions for porous MoN, TiN and Ta₅N₆ single crystals. (b,d,f) The high-resolution Cs-corrected STEM pictures of porous MoN, TiN and Ta₅N₆ single crystals.

We further summarize the relationship between coordination structure, surface area, gravimetric capacitance and areal capacitance for porous MoN, TiN and Ta₅N₆ single crystals as shown in **Supplementary Figure 20**. The porous MoN single crystal shows the highest gravimetric capacitance of ~2000 Fg⁻¹ in alkaline electrolyte and ~800 Fg⁻¹ in acidic electrolyte. And the area capacitance of ~8 F cm⁻² in alkaline electrolyte and ~5 F cm⁻² in acidic electrolyte are also obtained.

Supplementary Figure 20. The correlation between the unsaturated coordination structure, specific area, area of sample, gravimetric capacitance and areal capacitance for porous MoN, TiN and Ta₅N₆ single crystals with the thickness of 70 μm .

6. The nearly rectangular CV curves and the triangular GCD curves indicated the good capacitance property of MoN. In certain extent, such behavior looks like the EDLC of carbon materials. Thus, it is strongly recommended to study the real charge storage mechanism in depth.

Answer: Thank you very much for your comments and we further conduct supplementary experiment to confirm the pseudocapacitance with chemical adsorption. It is true that the EDLC of carbon materials normally show the rectangular CV curves and triangular GCD curves which are similar to that of pseudocapacitance with chemical adsorption. However, the obvious difference between the EDLC and pseudocapacitance is that the value of normalized specific area capacitance (areal capacitance divide by the BET surface area) of carbon materials is much lower than pseudocapacitance. The EDLC of carbon materials is mainly dominated by the physical adsorption which is much lower than the pseudocapacitance of current materials dominated by strong chemical adsorption.

We further grow porous MoN single crystal with the thickness of 18 and 36 μm and measure the specific area capacitance normalized relative to their BET surface areas. We get a very high normalized area capacitance value of more than 3500 $\mu\text{F cm}^{-2}$ which is among the highest values of current materials that exhibit pseudocapacitance and hundreds to thousands times larger than the value of the EDLC of carbon materials (*Nat Mater*, 2017, 16, 220). The pseudocapacitance is mainly dominated by the surface chemical adsorption on nitride surfaces. We further grow the porous MoN single crystal with the thickness ranging to 500 μm and investigate the correlation of pseudocapacitance with the electrode thickness. As shown in **Supplementary Figure 19**, the effective chemisorption takes place below the thickness of $\sim 70 \mu\text{m}$ in the porous microstructures which delivers the pseudocapacitance for porous single crystals. Just because of the effective thickness, the highest specific area capacitance is observed for the porous MoN single crystals with the thickness of 70 μm .

Supplementary Figure 19. (a-g) SEM image for the porous MoN single crystals with different thickness grown on the PbMoO₄ substrates with the area of 1 × 1 cm². (h) CV curves of porous MoN single crystals with different thickness at scan rate of 5 mVs⁻¹. (i) GCD curves of porous MoN single crystals with different thickness at the current density of 2 mA cm⁻². (j) The dependence of areal capacitance on the thickness of porous MoN single crystals at the scan rate of 2 mVs⁻¹. (k) The dependence of specific gravimetric capacitance on the thickness of porous MoN single crystals. (l) The dependence of normalized BET areal capacitance on the thickness of porous MoN single crystals. (m) BET surface areas of 1 × 1 cm² single crystals with different thickness grown on the PbMoO₄ substrates. (n) BET specific areas of the porous MoN single crystals with different thickness grown on the PbMoO₄ substrates. (o) The dependence of average pore size with different thick thickness of porous MoN single crystals.

7. The authors state that “The metallic porous MoN single crystal reaches a very high value of 8.8 Fcm⁻² at current density of 5 mA cm⁻² in a CD measurement, which represents the highest pseudocapacitance and is ~3-10 times higher than reported values with oxide and nitride electrodes”. I found that the main reason is that they used the PbMoO₄ substrates (20 mm × 10 mm × 0.5 mm). That to say, the very thick electrode (might be about 500 μm) resulted in the high areal capacitance shown in this paper, instead of the intrinsic characteristic MoN. So, the

comparison listed in the Figure 5f is not fully fair competition.

Answer: Thank you very much for your comments and we further conduct supplementary experiments. We fully agree with the reviewer on the influence of electrode thickness on the pseudocapacitance performance in our work. We then grow a series of porous MoN single crystals with thickness ranging from 18 to 500 μm as shown in **Supplementary Figure 19(a-g)** and investigate the correlation of pseudocapacitance with the electrode thickness. According to our BET measurements, the surface areas linearly increases with the electrode thickness which indicates the homogeneous distribution of pores at the average size of 40-50 nm in the microstructures as shown in **Supplementary Figure 19 (m) and (o)**.

We further measure the pseudocapacitance of porous MoN single crystals in relation to the electrode thickness. It is observed that the effective chemisorption takes place below the thickness of $\sim 70 \mu\text{m}$ in the porous microstructures, which leads to the pseudocapacitance with porous MoN single crystals. As shown in **Supplementary Figure 19 (j)**, the pseudocapacitance slightly increases with thickness ranging from 18 to 70 μm but generally remain unchanged even though the electrode thickness finally increase to 500 μm . For the porous MoN electrode with thickness of 70 μm , the specific gravimetric capacitance reaches as high as 772 F g^{-1} . And the gravimetric capacitance are as high as 1692 F g^{-1} and 2031 F g^{-1} and for the porous MoN single crystals with the thickness of 36 and 18 μm as shown in **Supplementary Figure 19 (k)**, respectively, which is mainly dominated by the intrinsic characteristic of MoN and much higher than most of the currently reported materials.

8. I cannot find the solid data to support the excellent rate performance the authors mentioned in this paper.

Answer: Thank you very much for your comments. We fully agree with the reviewer and we have revised the expression in revision.

9. What is the deep reason that porous MoN showed excellent cycling stability?

Answer: Thank you very much for your comments and we have provided the explanation in revision. We believe that there may be 2 reasons for the excellent cycling stability. Firstly, these transition metal nitrides are chemically stable both in acid and alkali solutions even in the presence of chemical adsorption of H^+ or OH^- on surfaces. Secondly, these porous single crystals, free of grain boundary, have the unique advantage of structural coherence, which reduces the interface defects at the largest extent and hence enhances the stability.

Reviewer #3 (Remarks to the Author):

In this work, the authors developed a series porous metal nitride as electrode for SCs. The obtained material showed attractive performance when applied for SCs. However, this paper do not provide some remarkable scientific viewpoints. In some way, the characterization supported

points in this work. I recommend this paper for publication in Nature Communications after major revision.

Answer: Thank you very much for your comments. We have fully conducted supplementary experiments and carefully revised the manuscript in revision. In our work, we grow metallic porous nitride single crystals at 2 cm scale to combine the advantages of large surface area with porous microstructure, high conductivity with metallic states and ordered active sites with unsaturated coordination. The long-range ordering of resolved local surface defect structures of transition metals with unsaturated coordination account for the fast redox reactions in chemisorption while the high conductivity together with porous microstructure facilitate the charge transfer and species diffusion in electrodes.

1. The authors claimed they prepared metal nitride single crystals such as MoN, Ta₅N₆ and TiN, however, the normal XRD test can't confirm the single crystal quality, the FWHM and X- ray rocking curve should be given.

Answer: Thank you very much for your comments. Yes, we agree with the reviewer on the approach of rocking curve to analyze the single crystallinity. We further conduct the measurement of rocking curve curves and the FWHM is as small as 0.179, 0.193 and 0.092° for the porous MoN, Ta₅N₆ and TiN single crystals, respectively, as shown in **Supplementary Figure 1**.

Supplementary Figure 1. The rocking curve curves of the porous single crystals. The FWHM is 0.179, 0.193 and 0.092° for the porous MoN, Ta₅N₆ and TiN single crystals, respectively.

2. In this manuscript, the authors consider the unsaturated metal-nitrogen active sites benefits for the pseudocapacitance, those active sites in what way absorb the electrolyte ions, what kinds of redox reaction takes place? Some characterization to prove the unsaturated metal-nitrogen should be given, like EPR.

Answer: Thank you very much for your comments. We further conduct Electron Paramagnetic Resonance (EPR) to analyze the porous MoN single crystal which shows EPR signals with g factor of 2.27 that arises from free electron in coordinatively-unsaturated sites. We use spherical aberration corrected scanning transmission electron microscope (Cs-corrected STEM) coupled with focused ion beam (FIB) investigate the coordination structures of the porous single crystal.

Supplementary Figure 12. The coordination structures for the porous MoN single crystal. The long range ordering of active sites would give rise to electrochemically active surface and the ordered active sites would accommodate the redox reactions on surfaces. **a** The coordination structures for the porous MoN single crystal. **b** HS-LEIS spectrum with He⁺ (3 keV) and Ne⁺ (5 keV) ions sources. **c** Cs-corrected STEM image of the MoN single crystal. **d** Simulated charge density graph of the outmost atomic surface layer of single-crystalline MoN. The dark-blue and light-blue balls represent Mo and N atoms, respectively. **e** OH⁻ adsorption on MoN surface.

We use high-sensitive low energy ion scattering (HS-LEIS) with He⁺ (3 keV) and Ne⁺ (5 keV) ions sources to confirm the outmost atomic surface of porous single crystals. We use DFT calculation to disclose the adsorption energy of H⁺ and OH⁻ on the MoN surface. We then summarize the unsaturated metal-nitrogen active sites and adsorption configuration as shown in **Supplementary Figure 12**. The metal-nitrogen coordination structure would function as active sites to accommodate the fast redox reactions in chemisorption on electrode surfaces.

3. The author claimed microstructures of porous MoN in the paper, how to define? As an important characterization method, the specific surface area and pore size distribution should be given.

Answer: Thank you very much for your comments and we further conduct BET measurements to analyze the microstructures. The specific surface areas are $\sim 14 \text{ m}^2 \text{ g}^{-1}$ with the average pore size of $\sim 40 \text{ nm}$ for porous MoN single crystals as shown in **Supplementary Figure 6**, which indicates the homogeneous distribution of porous microstructures even with different thickness.

Supplementary Figure 6. The surface specific areas and average pore sizes for the porous MoN, Ta₅N₆ and TiN single crystals. **a** The surface specific area. **b** Average pore sizes for the porous MoN, Ta₅N₆ and TiN single crystals. The error bar indicates the standard deviation in repeated measurements.

4. The authors think they obtain the highest areal pseudocapacitance of 8.8 Fcm^{-2} for the MoN electrode, however, I suggest they should compare more recent articles when they say the highest capacitance, such as the Ni₃S₂ can achieve $21.54/9.21 \text{ Fcm}^{-2}$ at $2/5 \text{ mA cm}^{-2}$ (J. Mater. Chem. A, 2018, 6, 22115); the NiCo₂S₄ obtain the areal capacitance of 14.39 F cm^{-2} at 5 mA cm^{-2} (J. Power Sources, 2014, 254, 249); and the capacitance of the MnO₂ is 44.13 F cm^{-2} . (Joule, 2019, 3, 459).

Answer: Thank you very much for your comments. We have cited the references in revision. We further conduct supplementary experiments using the porous MoN single crystals with different thickness. We can see that the effective chemisorption takes place below the thickness of $\sim 70 \mu\text{m}$ in the porous microstructures, which leads to the pseudocapacitance with porous MoN single crystals. As shown in **Supplementary Figure 19(j)**, the pseudocapacitance increases with thickness ranging from 18 to $70 \mu\text{m}$ but generally remain unchanged even though the electrode thickness finally increases to $500 \mu\text{m}$. We therefore conclude that the effective thickness is $\sim 70 \mu\text{m}$ for the porous MoN single crystals.

We further conduct the measurement of the specific area capacitance with porous MoN single crystal with the thickness of 18- $500 \mu\text{m}$ as shown in **Supplementary Figure 19**. We get a very high specific gravimetric capacitance of 2061 Fg^{-1} for porous MoN single crystal with the thickness of $18 \mu\text{m}$, which is among the highest values of current materials. We have also calculated the normalized specific area capacitance by dividing the capacitance with BET surface area. We obtain a very high value of normalized specific area capacitance of more than $3500 \mu\text{F cm}^{-2}$ for the porous MoN single crystals with the thickness below $70 \mu\text{m}$, which is also among the highest values of current materials. These enhanced pseudocapacitance mainly arises from the intrinsic property of the metallic porous MoN single crystals.

Supplementary Figure 19. (a-g) SEM image for the porous MoN single crystals with different thickness grown on the PbMoO_4 substrates with the area of $1 \times 1 \text{ cm}^2$. (h) CV curves of porous MoN single crystals with different thickness at scan rate of 5 mVs^{-1} . (i) GCD curves of porous MoN single crystals with different thickness at the current density of 2 mA cm^{-2} . (j) The dependence of areal capacitance on the thickness of porous MoN single crystals at the scan rate of 2 mVs^{-1} . (k) The dependence of specific gravimetric capacitance on the thickness of porous MoN single crystals. (l) The dependence of normalized BET areal capacitance on the thickness of porous MoN single crystals. (m) BET surface areas of $1 \times 1 \text{ cm}^2$ single crystals with different thickness grown on the PbMoO_4 substrates. (n) BET specific areas of the porous MoN single crystals with different thickness grown on the PbMoO_4 substrates. (o) The dependence of average pore size with different thick thickness of porous MoN single crystals.

5. The conductivity of the MoN, TiN and Ta_5N_6 single crystals are 0.77×10^4 , 0.45×10^4 and $0.8 \times 10^4 \text{ Scm}^{-1}$, how did you get these data, references or measurement?

Answer: Thank you very much for your comments and we provide the measured data lines in

revision. As shown in **Supplementary Figure 11**, all these three porous single crystals demonstrate metallic conduction behaviors with the highest conductivity of $\sim 1\text{-}3 \times 10^4 \text{ S cm}^{-1}$ observed at the temperatures of $\sim 50 \text{ K}$ while the conductivity is decreased by $\sim 50\%$ at room temperature.

Supplementary Figure 11. The relationship between conductivity and temperature for the porous MoN, Ta₅N₆ and TiN single crystals. The porous single crystals demonstrate metallic conduction behaviors.

6. The adsorption energy from the theoretical calculation is inappropriate, from the text, the adsorption energy of H and OH on (001) Ta₅N₆ surface are -1.10~-1.32 eV and -2.19~-2.56 eV, respectively, so the higher adsorption energy should be OH, however, it is not consistent with the experimental result.

Answer: Thank you very much for your comments. We fully understand the reviewer's concern on the higher energy of chemisorption in alkaline electrolyte solution. Actually, the chemisorption is only a necessary condition while the higher adsorption energy indicates the higher energy barrier of desorption on surface. Reasonable chemisorption energy would be favorable to the enhanced pseudocapacitance with porous single crystals.

7. The MoN and TiN surface deliver higher pseudocapacitance in alkaline electrolyte, and why the Ta₅N₆ obtain the higher capacitance in acidic electrolyte? Please give a reasonable explanation?

Answer: Thank you very much for your comments. We use DFT calculation to understand the chemisorption energy on Ta₅N₆ surface both in acidic and alkaline electrolyte solution. The chemisorption of OH⁻ is as high as -2.19~-2.56 eV which is much larger than that the chemisorption energy of H⁺ on Ta₅N₆ surface. The higher adsorption energy indicates the higher energy barrier of desorption on surface which would also limit the reversible redox reactions on porous nitride surfaces. Reasonable chemisorption energy would be favorable to the enhanced pseudocapacitance with porous single crystals.

8. The voltage windows of CV and GCD tests are different (Figure 5a, 5b and 5d), how does the author choose the right voltage range?

Answer: Thank you very much for your comments. We fully agree with the reviewer and we are sorry for this technical mistake. Actually the voltage range are 0-0.5 V vs RHE in all CV and GCD tests and we have corrected all these errors in the revised manuscript and supplementary materials.

9. In Figure 5c EIS, the unit on the Y-axis should $-Z''$, the sentence “the high-frequency region indicates very fast species diffusion in MoN single crystal” is a conceptual mistake.

Answer: Thank you very much for your comments and we have corrected this conceptual mistake in revision. We have revised “ Z'' ” to “ $-Z''$ ” in EIS in Figure 5c and Supplementary Figures. We have revised the sentence “the high-frequency region indicates very fast species diffusion in MoN single crystal” to “the high-frequency region indicates a very low resistivity of MoN single crystal in aqueous alkali solution”.

REVIEWERS' COMMENTS:

Reviewer #1 (Remarks to the Author):

Authors have revised the manuscript reasonably well to be accepted in Nature Communications

Reviewer #2 (Remarks to the Author):

After reviewing the responses to my comments I trust that the authors did a very good work on addressing all the issues that were raised during the analysis of the first version of the manuscript. My comments were addressed satisfactorily and the revised manuscript is sufficiently improved to recommend its acceptance to Nature Communications.

Reviewer #3 (Remarks to the Author):

The manuscript is well revised based on the reviewer's comments except for following points.

1. The Figure 5a is an unmodified data in the revised manuscript and it should be corrected.
2. The authors claim the g factor of the porous MoN single crystal is 2.27, however, the relevant EPR test data should be given in the main text or supporting information.

Response to reviewers

Reviewer #1 (Remarks to the Author):

Authors have revised the manuscript reasonably well to be accepted in Nature Communications

Answer: Thank you very much for your comments.

Reviewer #2 (Remarks to the Author):

After reviewing the responses to my comments I trust that the authors did a very good work on addressing all the issues that were raised during the analysis of the first version of the manuscript. My comments were addressed satisfactorily and the revised manuscript is sufficiently improved to recommend its acceptance to Nature Communications.

Answer: Thank you very much for your comments.

Reviewer #3 (Remarks to the Author):

The manuscript is well revised based on the reviewer's comments except for following points.

Answer: Thank you very much. We have made further revision according to your comments.

1. The Figure 5a is an unmodified data in the revised manuscript and it should be corrected.

Answer: Thank you very much for your comments. We have corrected Figure 5a in revision.

Figure 5a. CV curves at different scan rates of porous MoN single crystal.

2. The authors claim the g factor of the porous MoN single crystal is 2.27, however, the relevant EPR test data should be given in the main text or supporting information.

Answer: Thank you very much. We have shown the ESR in Supplementary Figure 12e.

Supplementary Fig.12e. The ESR of MoN single crystal.